# Explaining Defects of the Universal Vacua with Black Holes-Hedgehogs and Strings

**C. R. Das [1,*,†]** , **L. V. Laperashvili [2,†]**, **H. B. Nielsen [3,†]** and **B. G. Sidharth [4,†]**

1   Bogoliubov Laboratory of Theoretical Physics, Joint Institute for Nuclear Research, Joliot-Curie 6, 141980 Dubna, Moscow region, Russia

2   The Institute of Theoretical and Experimental Physics, National Research Center "Kurchatov Institute", Bolshaya Cheremushkinskaya, 25, 117218 Moscow, Russia; laper@itep.ru

3   Niels Bohr Institute, Blegdamsvej, 17-21, DK 2100 Copenhagen, Denmark; hbech@nbi.dk

4   International Institute of Applicable Mathematics and Information Sciences, B.M. Birla Science Centre, Adarsh Nagar, Hyderabad 500063, India; birlasc@gmail.com

*   Correspondence: das@theor.jinr.ru; Tel.: +7-496-216-3047

†   These authors contributed equally to this work.

**Abstract:** Assuming the Multiple Point Principle (MPP) as a new law of Nature, we considered the existence of the two degenerate vacua of the Universe: (a) the first Electroweak (EW) vacuum at $v_1 \approx 246$ GeV—"true vacuum", and (b) the second Planck scale "false vacuum" at $v_2 \sim 10^{18}$ GeV. In these vacua, we investigated different topological defects. The main aim of the paper is an investigation of the black-hole-hedgehogs configurations as defects of the false vacuum. In the framework of the $f(R)$ gravity, described by the Gravi-Weak unification model, we considered a black-hole solution, which corresponds to a "hedgehog"—global monopole, that has been "swallowed" by the black-hole with mass core $M_{BH} \sim 10^{18}$ GeV and radius $\delta \sim 10^{-21}$ GeV$^{-1}$. Considering the results of the hedgehog lattice theory in the framework of the $SU(2)$ Yang-Mills gauge-invariant theory with hedgehogs in the Wilson loops, we have used the critical value of temperature for the hedgehogs' confinement phase ($T_c \sim 10^{18}$ GeV). This result gave us the possibility to conclude that the SM shows a new physics (with contributions of the $SU(2)$-triplet Higgs bosons) at the scale $\sim 10$ TeV. This theory predicts the stability of the EW-vacuum and the accuracy of the MPP.

**Keywords:** hedgehogs; topological defects; multiple point principle

**PACS:** 04.50.Kd; 12.10.-g; 95.35.+d; 95.36.+x; 98.80.Es; 98.80.Cq; 12.60.-i; 11.10.Nx

**MSC:** 83E99

## 1. Introduction

The present review is devoted to studying topological defects of the universal vacua.

During the expansion after the Planck era, the early Universe underwent a series of phase transitions as a result of which there were arisen such vacuum topological defects (widely discussed in literature) as monopoles or hedgehogs (point defects), strings (line defects), bubbles and domain walls (sheet defects). These topological defects appeared due to the breakdown of local or global gauge symmetries.

This paper is essentially based on the discovery that a cosmological constant of our Universe is extremely small, almost zero [1–5]. We considered a Multiple Point Principle (MPP) first suggested by D.L. Bennett and H.B. Nielsen [6], which predicts the existence in Nature of several degenerate vacua with a very small energy density (cosmological constants).

The model developed in this article confirms the existence of the two degenerate vacua of the Universe: The first ("true") Electroweak (EW) vacuum with VEV $v_1 \approx 246$ GeV, and the second ("false") Planck scale vacuum with VEV $v_2 \sim 10^{18}$ GeV.

The main idea of this paper is the investigation of hedgehog's configurations [7,8] as defects of the false vacuum. We have shown that at super high (Planck scale) energies the black-holes-hedgehogs are responsible for the creation of the false vacuum of the Universe. In the framework of the $f(R)$ gravity, we have obtained a solution for a global monopole, which is a black-hole-hedgehog at the Planck scale. Here we have used the $f(R)$ gravity predicted by the Gravi-Weak unification model previously developed by authors in papers [9–12].

Using the results of Refs. [13,14] obtained for the $SU(2)$ Yang-Mills theory of the gauge-invariant hedgehog-like structures in the Wilson loops, we have considered the lattice theory giving the critical value of temperature for the hedgehogs' confinement phase. Considering the hedgehog lattice theory, we have concluded that hedgehogs can exist only at the energy scale $\mu \gtrsim 10^4$ GeV. Triplet Higgs fields $\Phi^a$ (with $a = 1, 2, 3$), which are responsible for the formation of hedgehogs, can show a new physics at the scale $\sim 10$ TeV.

In Section 2 we reviewed the Multiple Point Principle (MPP) suggested by D.L. Bennett and H.B. Nielsen [6]. In the assumption of the existence of the two degenerate vacua (Electroweak vacuum at $v_1 \approx 246$ GeV, and Planck scale one at $v_2 \sim 10^{18}$ GeV), Froggatt and Nielsen [15] obtained the first prediction of the top quark and Higgs boson masses, which was further improved by several authors in the next approximations. Section 3 is devoted to the general properties of topological defects of the universal vacua. We considered topological defects in the "false vacuum", which is presented as a spherical bubble spontaneously produced in the de-Sitter like Universe. The space-time inside the bubble, which we refer to as a "true vacuum", has the geometry of an open Friedmann-Lemaitre-Robertson-Walker (FLRW) Universe. Section 4 is devoted to the Gravi-Weak unification (GWU) model [9–12] as an example of the $f(R)$ gravity. Section 4.1 considers the existence of the de-Sitter solutions in the Planck phase. Section 4.2 is devoted to calculations of parameters of the GWU-model, where we predicted the Planck scale false vacuum VEV equal to $v_2 \approx 6.28 \times 10^{18}$ GeV. In Section 5 we have investigated the hedgehog's configurations as defects of the false vacuum. We obtained a solution for a black-hole in the framework of the $f(R)$ gravity, which corresponds to a global monopole "swallowed" by a black-hole. The metric around of the global monopole was considered in Section 5.1. The mass $M_{BH}$, radius $\delta$ and "horizon radius" $r_h$ of the black-hole-hedgehog were estimated in Section 5.2. Section 6 is devoted to the lattice-like structure of the false vacuum which is described by a non-differentiable space-time: by a foam of black-holes, having lattice-like structure, in which sites are black-holes with "hedgehog" monopoles inside them. This manifold is described by a non-commutative geometry predicted an almost zero cosmological constant. The phase transition from the "false vacuum" to the "true vacuum" was considered in Section 7, where it was shown that the Electroweak spontaneous breakdown of symmetry $SU(2)_L \times U(1)_Y \rightarrow U(1)_{el.mag}$ created new topological defects of EW vacuum: the Abrikosov-Nielsen-Olesen closed magnetic vortices ("ANO strings") of the Abelian Higgs model and Sidharth's Compton phase objects. Then the "true vacuum" (EW-vacuum) again presents the non-differentiable manifold with non-commutative geometry and again has an almost zero cosmological constant. Here we estimated the black-hole-hedgehog's mass and radius: $M_{BH} \approx 3.65 \times 10^{18}$ GeV and $\delta \approx 0.29\lambda_{Pl} \approx 10^{-21}$ GeV$^{-1}$ near the second vacuum $v_2$. In Section 7.1 we emphasize that due to the energy conservation law, the vacuum density before the phase transition is equal to the vacuum density after the phase transition, and we have

$$\rho_{vac}(\text{at Planck scale}) = \rho_{vac}(\text{at EW scale}).$$

Therefore, we confirmed the Multiple Point Principle: we have two degenerate vacua $v_1$ and $v_2$ with an almost zero vacuum energy (cosmological constants). This means that our EW-vacuum, in which we live, is stable. The Planck scale vacuum cannot be negative: $V_{eff}(min_1) = V_{eff}(min_2)$, these potentials are equal exactly. In Section 8 hedgehogs in Wilson loops of the $SU(2)$ Yang-Mills

theory, and phase transitions in this theory were investigated using the results of Refs. [13,14]. Their lattice results gave the critical value of the temperature for the hedgehog's confinement phase: $\beta_{crit} \approx 2.5$, and this result gives the value of critical temperature $T_c \sim 10^{18}$ GeV. In Section 9 we show that the hedgehog's confinement happens at energy $\sim 10$ TeV, which is a threshold energy of the production of a pair of the $SU(2)$-triplet Higgs bosons. In Section 10 we reviewed the problem of the vacuum stability (for example see Refs. [16]) in the Standard Model. In Section 11 we show that hedgehogs can contribute at energy scale $\mu > 10^4$ GeV. Therefore, a triplet Higgs field $\Phi^a$ provides a new physics at the scale $\sim 10$ TeV. In this Section 11, we predict an exact stability of the EW-vacuum and the accuracy of the MPP.

## 2. Degenerate Vacua of the Universe

This paper is based on the new law of Nature named Multiple Point Principle (MPP) which was suggested by D.L. Bennett and H.B. Nielsen in Ref. [6]. The MPP means: There exist in Nature several degenerate vacua with very small energy density or cosmological constants.

Vacuum energy density of our Universe is the Dark Energy (DE). It is related with cosmological constant $\Lambda$ by the following way:

$$\rho_{DE} = \rho_{vac} = (M_{Pl}^{red})^2 \Lambda, \tag{1}$$

where $M_{Pl}^{red}$ is the reduced Planck mass: $M_{Pl}^{red} \simeq 2.43 \times 10^{18}$ GeV. At present, cosmological measurements give:

$$\rho_{DE} \simeq (2 \times 10^{-3} \text{ eV})^4, \tag{2}$$

which means a tiny value of the cosmological constant:

$$\Lambda \simeq 10^{-84} \text{ GeV}^2. \tag{3}$$

This tiny value of $\rho_{DE}$ was first predicted by B.G. Sidharth in 1997 year [3,4]. In the 1998 year S. Perlmutter, B. Schmidt and A. Riess [5] were awarded the Nobel Prize for the discovery of the accelerating expansion of the Universe.

Having an extremely small cosmological constant of our Universe, Bennett, Froggatt and Nielsen [6,15,17,18] assumed to consider only zero, or almost zero, cosmological constants for all vacua existing in Nature.

The MPP theory was developed in a lot of papers by H.B. Nielsen and his collaborators (see for example, Refs. [6,15,17–32] and recent Refs. [33–37] by other authors).

Restricted to pure Standard Model (SM) we assumed the existence of only three vacua:

1. Present Electroweak vacuum, "true vacuum", in which we live.
   It has vacuum expectation value (VEV) of the Higgs field equal to:

$$v_1 = \langle \phi_H \rangle \approx 246 \text{ GeV}. \tag{4}$$

2. High Higgs field vacuum, "false vacuum"—Planck scale vacuum, which has the following VEV:

$$v_2 = \langle \phi_H \rangle \sim 10^{18} \text{ GeV}. \tag{5}$$

3. Condensate vacuum. This third vacuum is a very speculative possible state inside the pure SM, which contains a lot of strongly bound states, each bound from 6 top + 6 anti-top quarks (see Refs. [38–42]).

From experimental results for these three vacua, cosmological constants—minima of the Higgs effective potentials $V_{eff}(\phi_H)$—are not exactly equal to zero. Nevertheless, they are extremely small. For this reason, Bennett, Froggatt and Nielsen assumed to consider zero cosmological constants as a

good approximation. Then according to the MPP, we have a model of pure SM being fine-tuned in such a way that these three vacua proposed have just zero energy density.

If the effective potential has three degenerate minima, then the following requirements are satisfied:

$$V_{eff}(\phi^2_{min1}) = V_{eff}(\phi^2_{min2}) = V_{eff}(\phi^2_{min3}) = 0, \tag{6}$$

and

$$V'_{eff}(\phi^2_{min1}) = V'_{eff}(\phi^2_{min2}) = V'_{eff}(\phi^2_{min3}) = 0, \tag{7}$$

where

$$V'(\phi^2) = \frac{\partial V}{\partial \phi^2}. \tag{8}$$

Here we assume that:

$$V_{eff}(\phi^2_{min1}) = V_{present}, \quad V_{eff}(\phi^2_{min2}) = V_{high\ field}, \quad \text{and} \quad V_{eff}(\phi^2_{min3}) = V_{condensate}. \tag{9}$$

Assuming the existence of the two degenerate vacua in the SM:

a. the first Electroweak vacuum at $v_1 \approx 246$ GeV, and
b. the second Planck scale vacuum at $v_2 \sim 10^{18}$ GeV,

Froggatt and Nielsen predicted in Ref. [15] the top-quark and Higgs boson masses:

$$M_t = 173 \pm 5 \text{ GeV}; \quad M_H = 135 \pm 10 \text{ GeV}. \tag{10}$$

In Figure 1 it is shown the existence of the second (non-standard) minimum of the effective Higgs potential in the pure SM at the Planck scale.

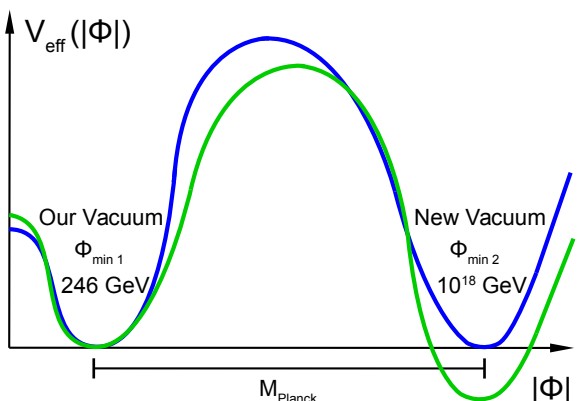

**Figure 1.** Minima of the effective Higgs potential in the pure Standard Model, which correspond to the first Electroweak "true vacuum", and to the second Planck scale "false vacuum".

## 3. Topological Defects of the Universal Vacua

Topological structures in fields are as important as the fields themselves. The presence of defects determines the special features of the vacuum.

It is well known that in the early Universe topological defects may be created in the vacuum during the vacuum phase transitions. The early Universe underwent a series of phase transitions, each one spontaneously breaking some symmetry in particle physics and giving rise to topological defects of some kind, which can play an essential role throughout the subsequent evolution of the Universe.

In the context of the General Relativity, Barriola and Vilenkin (see Ref. [43]) studied the gravitational effects of a global monopole as a spherically symmetric topological defect. The authors found, that the gravitational effect of the global monopole is repulsive in nature. Thus, one may

expect that the global monopole and cosmological constants are connected through their common manifestation as the origin of repulsive gravity. Moreover, both the cosmological constant and vacuum expectation value (VEV) are connected while the VEV is connected to the topological defects. All these points lead us to a simple conjecture: There must be a common connection among the cosmological constant, topological defects and the vacuum expectation values (VEVs).

Different phase transitions have resulted, during the expansion of the early Universe after the Planck era. They produced the formation of the various kind of topological defects: point defects (monopoles, hedgehogs, etc.); line defects (strings, vortices), and sheet defects (for example, domain walls). The topology of the vacuum manifold dictates the nature of these topological defects, appearing due to the breakdown of local or global gauge symmetries.

In the present paper, we shall discuss another potentially observable manifestation of topological defects. It has been shown in Ref. [44] that topological defects, like spherical domain walls and circular loops of cosmic string, can be spontaneously produced in a de-Sitter like Universe. At the moment of creation of the new Universe by the new vacuum or topological structure giving rise to the initial radii of walls and strings are close to the de-Sitter horizon. This horizon corresponds to a radius today of the order:

$$R_{un} \simeq R_{de\text{-}Sitter\ horizon} \simeq 10^{28}\ \text{cm}. \tag{11}$$

In the present paper, we study the evolution of the two bubbles: one having a "false vacuum", and the other one having a "true vacuum". The bubble, which we shall refer to as the false vacuum, to be a de-Sitter space with a constant expansion rate $H_F$. It is convenient to use flat de-Sitter coordinates to describe the background of the inflating false vacuum:

$$ds^2 = dt^2 - e^{2H_F t}(dr^2 + r^2 d\Omega^2), \tag{12}$$

where

$$d\Omega^2 = d\theta^2 + \sin^2 \theta d\phi^2. \tag{13}$$

The space-time inside the bubble, which we shall refer to as a true vacuum, has the geometry of an open Friedmann-Lemaitre-Robertson-Walker (FLRW) Universe (see for example review [45]):

$$ds^2 = d\tau^2 - a(\tau)^2 (d\xi^2 + \sinh^2 \xi d\Omega^2), \tag{14}$$

where $a(\tau)$ is a scale factor with cosmic time $\tau$. In the true vacuum, we have a constant expansion rate $H_T$, which has the meaning of the slow-roll inflation rate inside the bubble at the early stage of its evolution.

Cosmological theory of bubbles was developed in a lot of papers by A. Vilenkin and his collaborators (see for example Refs. [44,46,47]). The physical properties of defects depend on the embedding vacuum.

## 4. Gravi-Weak Unification and Hedgehogs as Defects of the False Vacuum

In the paper [9] (using the ideas of Refs. [48,49]) we have considered a $Spin(4,4)$-group of the gravi-weak unification which is spontaneously broken into the $SL(2,C)^{(grav)} \times SU(2)^{(weak)}$. Such a model was constructed in agreement with experimental and astrophysical results. We assumed that after the Bing Bang there existed a Theory of the Everything (TOE) which rapidly was broken down to the direct product of the following gauge groups:

$$
\begin{aligned}
G_{(TOE)} \quad &\rightarrow \quad G_{(GW)} \times U(4) \rightarrow SL(2,C)^{(grav)} \times SU(2)^{(weak)} \times U(4) \\
&\rightarrow \quad SL(2,C)^{(grav)} \times SU(2)^{(weak)} \times SU(4) \times U(1)_Y \\
&\rightarrow \quad SL(2,C)^{(grav)} \times SU(2)^{(weak)} \times SU(3)_c \times U(1)_{(B-L)} \times U(1)_Y \\
&\rightarrow \quad SL(2,C)^{(grav)} \times SU(3)_c \times SU(2)_L \times U(1)_Y \times U(1)_{(B-L)} \\
&\rightarrow \quad SL(2,C)^{(grav)} \times G_{SM} \times U(1)_{(B-L)}.
\end{aligned}
$$

And below the see-saw scale ($M_R \sim 10^9 \sim 10^{14}$ GeV) we have the SM group of symmetry:

$$
G_{SM} = SU(3)_c \times SU(2)_L \times U(1)_Y.
$$

The action $S_{(GW)}$ of the Gravi-Weak unification (obtained in Ref. [9]) was given by the following expression:

$$
\begin{aligned}
S_{(GW)} \quad = \quad &-\frac{1}{g_{uni}} \int_{\mathfrak{M}} d^4x \sqrt{-g} \left[ \frac{1}{16} \left( R|\Phi|^2 - \frac{3}{2}|\Phi|^4 \right) \right. \\
&\left. + \frac{1}{16} \left( aR_{\mu\nu}R^{\mu\nu} + bR^2 \right) + \frac{1}{2}\mathcal{D}_\mu\Phi^\dagger \mathcal{D}^\mu\Phi + \frac{1}{4}F_{\mu\nu}^i F^{i\,\mu\nu} \right],
\end{aligned}
\tag{15}
$$

where $g_{uni}$ is a parameter of the graviweak unification, parameters $a, b$ (with $a + b = 1$) are "bare" coupling constants of the higher derivative gravity, $R$ is the Riemann curvature scalar, $R_{\mu\nu}$ is the Ricci tensor, $|\Phi|^2 = \Phi^a\Phi^a$ is a squared triplet Higgs field, where $\Phi^a$ (with $a = 1, 2, 3$) is an isovector scalar belonging to the adjoint representation of the $SU(2)$ gauge group of symmetry. In Equation (15):

$$
\mathcal{D}_\mu\Phi^a = \partial_\mu\Phi^a + g_2\epsilon^{abc}A_\mu^b\Phi^c
\tag{16}
$$

is a covariant derivative, and

$$
F_{\mu\nu}^a = \partial_\mu A_\nu^a - \partial_\nu A_\mu^a + g_2\epsilon^{abc}A_\mu^b A_\nu^c
\tag{17}
$$

is a curvature of the gauge field $A_\mu^a$ of the $SU(2)$ Yang-Mills theory. The coupling constant $g_2$ is a "bare" coupling constant of the $SU(2)$ weak interaction.

Some special case of the $f(R)$ gravity studied in [50–52], but in a general case of the $f(R)$ gravity, the action contains matter fields and can be presented by the following expression:

$$
S = \frac{1}{2\kappa} \int d^4x \sqrt{-g}\, f(R) + S_{grav} + S_{gauge} + S_m,
\tag{18}
$$

where $S_m$ corresponds to the part of the action associated with matter fields, fermions and Higgs fields.

Using the metric formalism, we obtain the following field equations:

$$
F(R)R_{\mu\nu} - \frac{1}{2}f(R)g_{\mu\nu} - \nabla_\mu\nabla_\nu F(R) + g_{\mu\nu}\Box F(R) = \kappa T_{\mu\nu}^m,
\tag{19}
$$

where:

$$
F(R) \equiv \frac{df(r)}{dr},
\tag{20}
$$

$\kappa = 8\pi G_N$, $G_N$ is the gravitational constant, and $T^m$ is the energy-momentum tensor derived from the matter action $S_m$.

*4.1. The Existence of the De-Sitter Solutions at the Early Time of the Universe*

It is well-known that at the early time of the Universe an acceleration era is described by the de-Sitter solutions (see for example [53,54]). The investigation of the problem that de-Sitter solutions

exist in the case of the action (15) was considered by authors of Ref. [48]. Our model [9] is a special case of the more general $SU(N)$ model [48], and we can assume that the Universe is inherently de-Sitter. Then the 4-spacetime is a hyperboloid in a 5-dimensional Minkowski space under the constraint:

$$x_0^2 + x_1^2 + x_2^2 + x_3^2 + x_4^2 = r_{dS}^2, \tag{21}$$

where $r_{dS}$ is a radius of a curvature of the de-Sitter space, or simply "the de-Sitter radius". The Hubble expansion of the Universe is then viewed as a process that approaches the asymptotic limit of a pure space which is de-Sitter in nature, evidenced that the cosmological constant $\Lambda$ describes the dark energy (DE) substance, which has become dominant in the Universe at later times:

$$\Omega_{DE} = \frac{\rho_{DE}}{\rho_{crit}} \simeq 0.75, \tag{22}$$

where $\rho_{DE}$ is the dark energy density and the critical density is:

$$\rho_{crit} = \frac{3H_0^2}{8\pi G_N} \simeq 1.88 \times 10^{-29}\, H_0^2, \tag{23}$$

where $H_0$ is the Hubble constant:

$$H_0 \simeq 1.5 \times 10^{-42}\ \text{GeV}. \tag{24}$$

Identifying the Einstein tensor as

$$G_{\mu\nu} = -\frac{3}{r_{dS}^2} g_{\mu\nu}, \tag{25}$$

we see that the only nontrivial component that satisfies this equation is a constant for the Ricci scalar:

$$R_0 = \frac{12}{r_{dS}^2}. \tag{26}$$

As it was shown in Ref. [48], the nontrivial vacuum solution to the action (15) is de-Sitter spacetime with a non-vanishing Higgs vacuum expectation value (VEV) of the triplet Higgs scalar field $\Phi$: $v_2 = \langle\Phi\rangle = \Phi_0$. The standard Higgs potential in Equation (15) has an extremum at $\Phi_0 = R/3$ (with $R > 0$), corresponding to a de-Sitter spacetime background solution:

$$R = R_0 = \frac{12}{r_{dS}^2} = 3v_2^2, \tag{27}$$

which implies vanishing curvature:

$$F_0 = \frac{1}{2}R_0 - \frac{1}{16}\Sigma_0 \Phi_0^2 \tag{28}$$

solving the field equations $DF = dF + [A, F] = 0$, and strictly minimizing the action (15).

Based on this picture, the origin of the cosmological constant (and DE) is associated with the inherent spacetime geometry, and not with vacuum energy of particles (we consider their contributions later). We note that as a fundamental constant under the de-Sitter symmetry, $r_{dS}$ is not a subject to quantum corrections. Local dynamics exist as fluctuations with respect to this cosmological background. In general, the de-Sitter space may be inherently unstable. The quantum instability of the de-Sitter space was investigated by various authors. Abbott and Deser [55] have shown that de-Sitter space is stable under a restricted class of classical gravitational perturbations. So any instability of the de-Sitter space may likely have a quantum origin. Ref. [56] demonstrated through the expectation value of the energy-momentum tensor for a system with a quantum field in a de-Sitter background space that in general, it contains a term that is proportional to the metric tensor and grows in time. As a result,

the curvature of the spacetime would decrease and the de-Sitter space tends to decay into the flat space (see Ref. [57]). The decay time of this process is of the order of the de-Sitter radius:

$$\tau \sim r_{dS} \simeq 1.33 \, H_0^{-1}. \tag{29}$$

Since the age of our Universe is smaller than $r_{dS}$, we are still observing the accelerating expansion in action.

Of course, we can consider the perturbation solutions of the de-Sitter solution but these perturbations are very small [53,54].

### 4.2. Parameters of the Gravi-Weak Unification Model

Assuming that at the first stage of the evolution (before the inflation), the Universe had the de-Sitter spacetime—maximally symmetric Lorentzian manifold with a constant and positive background scalar curvature $R$—we have obtained the following relations from the action (15):

(1) The vacuum expectation value $v_2$—the VEV of "the false vacuum"—is given by the de-Sitter scalar curvature $R$:

$$v_2^2 = \frac{R}{3}. \tag{30}$$

(2) At the Planck scale the squared coupling constant of the weak interaction is:

$$g_2^2 = g_{uni}. \tag{31}$$

The replacement:

$$\frac{\Phi^a}{g_2} \to \Phi^a \tag{32}$$

leads to the following GW-action:

$$
\begin{aligned}
S_{(GW)} \;=\; &-\int_{\mathfrak{M}} d^4x \sqrt{-g} \left( \frac{R}{16}|\Phi|^2 - \frac{3g_2^2}{32}|\Phi|^4 + \frac{1}{2}\mathcal{D}_\mu \Phi^\dagger \mathcal{D}^\mu \Phi + \frac{1}{4g_2^2} F^i_{\mu\nu} F^{i\,\mu\nu} \right. \\
&\left. + \text{grav. terms} \right),
\end{aligned} \tag{33}
$$

Now considering the VEV of the false vacuum as $v = v_2$, we have:

$$v^2 = \frac{R}{3g_2^2}. \tag{34}$$

The Einstein-Hilbert action of general relativity with the Einstein's cosmological constant $\Lambda_E$ is given by the following expression:

$$S_{EH} = -\frac{1}{\kappa} \int d^4x \sqrt{-g} \left( \frac{R}{2} - \Lambda_E \right). \tag{35}$$

(3) The comparison of the Lagrangian $L_{EH}$ with the Lagrangian given by Equation (33) near the false vacuum $v$ leads to the following relations for the Newton's gravitational constant $G_N$ and reduced Planck mass:

$$(M_{Pl}^{red})^2 = (8\pi G_N)^{-1} = \frac{1}{\kappa} = \frac{v^2}{8}. \tag{36}$$

(4) Then we have:

$$v = 2\sqrt{2} M_{Pl}^{red} \approx 6.28 \times 10^{18} \text{ GeV}, \tag{37}$$

and

$$\Lambda_E = \frac{3g_2^2}{4} v^2. \tag{38}$$

Equation (36) gives:

$$\frac{1}{\kappa}\Lambda_E = \frac{3g_2^2}{32}v^4. \tag{39}$$

Using the well-known in literature renormalization group equation (RGE) for the $SU(2)$ running constant $\alpha_2^{-1}(\mu)$, where $\alpha_2 = g_2^2/4\pi$ and $\mu$ is the energy scale, we can use the extrapolation of this value to the Planck scale [21,22] and obtain the following result:

$$\alpha_2(M_{Pl}) \sim \frac{1}{50}, \quad g_{uni} = g_2^2 = 4\pi\alpha_2(M_{Pl}) \approx 4\pi \times 0.02 \approx 0.25. \tag{40}$$

## 5. The Solution for the Black-Holes-Hedgehogs

A global monopole is described by the part $L_h$ of the Lagrangian $L_{(GW)}$ given by the action (33), which contains the $SU(2)$-triplet Higgs field $\Phi^a$, VEV of the second vacuum $v_2 = v$ and cosmological constant $\Lambda = \Lambda_E$:

$$
\begin{aligned}
L_h &= -\frac{R}{16}|\Phi|^2 + \frac{3g_2^2}{32}|\Phi|^4 - \frac{1}{2}\partial_\mu\Phi^a\partial^\mu\Phi^a + \Lambda_E \\
&= -\frac{1}{2}\partial_\mu\Phi^a\partial^\mu\Phi^a + \frac{\lambda}{4}\left(|\Phi|^2 - v^2\right)^2 + \frac{\Lambda_E}{\kappa} - \frac{\lambda}{4}v^4 \\
&= -\frac{1}{2}\partial_\mu\Phi^a\partial^\mu\Phi^a + \frac{\lambda}{4}\left(|\Phi|^2 - v^2\right)^2.
\end{aligned} \tag{41}
$$

Here we have:

$$\lambda = \frac{3g_2^2}{8}. \tag{42}$$

Substituting in Equation (42) the value $g_2^2 \approx 0.25$ given by Equation (40), we obtain:

$$\lambda \approx \frac{3}{32}. \tag{43}$$

Equation (39) gives:

$$\frac{\Lambda_E}{\kappa} = \frac{3g_2^2}{32}v^4 = \frac{\lambda}{4}v^4, \tag{44}$$

and in Equation (42) we have the compensation of the Einstein's cosmological term. Then

$$L_h = -\frac{1}{2}\partial_\mu\Phi^a\partial^\mu\Phi^a + V(\Phi), \tag{45}$$

where the Higgs potential is:

$$V(\Phi) = \frac{\lambda}{4}\left(|\Phi|^2 - v^2\right)^2. \tag{46}$$

This potential has a minimum at $\langle|\Phi|\rangle_{min} = v$, in which it vanishes:

$$V\left(|\Phi|_{min}^2\right) = V'\left(|\Phi|_{min}^2\right) = 0, \tag{47}$$

in agreement with the MPP conditions (6) and (7).

The field configurations describing a monopole-hedgehog [7,8] are:

$$
\begin{aligned}
\Phi^a &= vw(r)\frac{x^a}{r}, \\
A_\mu^a &= a(r)\epsilon_{\mu ab}\frac{x^b}{r},
\end{aligned} \tag{48}
$$

where $x^a x^a = r^2$ with $(a = 1, 2, 3)$, $w(r)$ and $a(r)$ are some structural functions. This solution is pointing radially. Here $\Phi^a$ is parallel to $\hat{r}$—the unit vector in the radial, and we have a "hedgehog" solution of Refs. [7,8]. The terminology "hedgehog" was first suggested by Alexander Polyakov in Ref. [8].

The field equations for $\Phi^a$ in the flat metric reduces to a single equation for $w(r)$:

$$w'' + \frac{2}{r}w' - \frac{2}{r^2}w - \frac{w(w^2 - 1)}{\delta^2} = 0, \tag{49}$$

where $\delta$ is the core radius of the hedgehog. In the flat space the hedgehog's core has the following size:

$$\delta \sim \frac{1}{\sqrt{\lambda}v}. \tag{50}$$

The function $w(r)$ grows linearly when $r < \delta$ and exponentially approaches unity as soon as $r > \delta$. Barriola and Vilenkin [43] took $w = 1$ outside the core which is an approximation to the exact solution. As a result, the functions $w(r)$ and $a(r)$ are constrained by the following conditions:

$$
\begin{aligned}
w(0) &= 0, \quad \text{and} \quad w(r) \to 1 \quad \text{when} \quad r \to \infty, \\
a(0) &= 0, \quad \text{and} \quad a(r) \sim -\frac{1}{r} \quad \text{when} \quad r \to \infty.
\end{aligned}
\tag{51}
$$

*5.1. The Metric Around of the Global Monopole*

The most general static metric around of the global monopole is a metric with spherical symmetry:

$$ds^2 = B(r)dt^2 - A(r)dr^2 - r^2(d\theta^2 + \sin^2\theta d\varphi^2). \tag{52}$$

For this metric the Ricci tensor has the following non-vanishing components:

$$
\begin{aligned}
R_{tt} &= -\frac{B''}{2A} + \frac{B'}{4A}\left(\frac{A'}{A} + \frac{B'}{B}\right) - \frac{1}{r}\frac{B'}{A}, \\
R_{rr} &= \frac{B''}{2B} + \frac{B'}{4B}\left(\frac{A'}{A} + \frac{B'}{B}\right) - \frac{1}{r}\frac{A'}{A}, \\
R_{\theta\theta} &= -1 + \frac{r}{2A}\left(-\frac{A'}{A} + \frac{B'}{B}\right) + \frac{1}{A}, \\
R_{\varphi\varphi} &= \sin^2\theta R_{\theta\theta}.
\end{aligned}
\tag{53}
$$

The energy-momentum tensor of the monopole is given by

$$
\begin{aligned}
T_t^t &= v^2\frac{w'^2}{2A} + v^2\frac{w^2}{r^2} + \frac{1}{4}\lambda v^4(w^2 - 1)^2, \\
T_r^r &= -v^2\frac{w'^2}{2A} + v^2\frac{w^2}{r^2} + \frac{1}{4}\lambda v^4(w^2 - 1)^2, \\
T_\theta^\theta = T_\varphi^\varphi &= v^2\frac{w'^2}{2A} + \frac{1}{4}\lambda v^4(w^2 - 1)^2.
\end{aligned}
\tag{54}
$$

Here $\kappa = 1$.

Considering the approximation used by Barriola and Vilenkin in Ref. [43], we obtain an approximate solution for monopole-hedgehog taking $w = 1$ out the core of the hedgehog (see also

Refs. [58–62]). In this case scalar curvature $R$ is constant and Equation (19) comes down to the Einstein's equation:

$$\frac{1}{A}\left(\frac{1}{r^2} - \frac{1}{r}\frac{A'}{A}\right) - \frac{1}{r^2} = \frac{1}{v^2}T_t^t, \tag{55}$$

$$\frac{1}{A}\left(\frac{1}{r^2} + \frac{1}{r}\frac{B'}{B}\right) - \frac{1}{r^2} = \frac{1}{v^2}T_r^r, \tag{56}$$

where the energy-momentum tensor is given by the following approximation:

$$T_t^t = T_r^r \approx \frac{v^2}{r^2},$$

$$T_\theta^\theta = T_\varphi^\varphi = 0. \tag{57}$$

Taking into account Equation (57), we obtain the following result by substraction of Equations (55) and (56):

$$\frac{A'}{A} + \frac{B'}{B} = 0, \tag{58}$$

and then asymptotically (when $r \to \infty$) we have:

$$A \approx B^{-1}. \tag{59}$$

From Equation (55) we obtain a general relation for the function $A(r)$:

$$A^{-1}(r) = 1 - \frac{1}{r}\int_0^r T_t^t\, r^2 dr. \tag{60}$$

In the limit $r \to \infty$ we obtain:

$$A(r) = 1 - \kappa v^2 - \frac{2G_N M}{r} + \dots$$

$$B(r) = \left(1 - \kappa v^2 - \frac{2G_N M}{r} + \dots\right)^{-1} \tag{61}$$

### 5.2. The Mass, Radius and Horizon Radius of the Black-Hole-Hedgehog

Equation (60) suggests the following equation for the hedgehog mass $M$:

$$M = 8\pi \int_0^\infty T_t^t\, r^2 dr, \tag{62}$$

or

$$M = 8\pi v^2 \int_0^\infty \left(w'^2 + \frac{w^2 - 1}{r^2} + \frac{(w^2 - 1)^2}{4\delta^2}\right) r^2 dr. \tag{63}$$

The function $w(r)$ was estimated in Ref. [61] at $r < \delta$:

$$w(r) \approx 1 - \exp\left(-\frac{r}{\delta}\right), \tag{64}$$

and we obtain an approximate value of the hedgehog mass:

$$M = M_{BH} \approx -8\pi v^2 \delta. \tag{65}$$

There is a repulsive gravitational potential due to this negative mass. A freely moving particle near the core of the black-hole experiences an outward proper acceleration:

$$\ddot{r} = -\frac{G_N M}{r} = \frac{G_N |M|}{r}. \tag{66}$$

We have obtained a global monopole with a huge mass (65), which has a property of the hedgehog. This is a black-hole solution, which corresponds to a global monopole-hedgehog that has been "swallowed" by a black-hole. Indeed, we have obtained the metric result by M. Barriola et al. [43] like:

$$ds^2 = \left(1 - \kappa v^2 + \frac{2G_N |M|}{r}\right) dt^2 - \frac{dr^2}{1 - \kappa v^2 + \frac{2G_N |M|}{r}} - r^2 \left(d\theta^2 + \sin^2\theta d\varphi^2\right). \tag{67}$$

A black-hole has a horizon. A horizon radius $r_h$ is found by solving the equation $A(r_h) = 0$:

$$1 - \kappa v^2 + \frac{2G_N |M|}{r_h} = 0, \tag{68}$$

and we have a solution:

$$r_h = \frac{2G_N |M|}{\kappa v^2 - 1}. \tag{69}$$

According to Equation (36), $\kappa v^2 = 8$, and we obtain the black-hole-hedgehog with a horizon radius:

$$r_h = \frac{2}{7} G_N |M| = \frac{2}{7} \times \frac{\kappa}{8\pi} \times |M| = \frac{2}{7} \times \frac{\kappa}{8\pi} \times 8\pi v^2 \delta \approx \frac{16}{7}\delta \approx 2.29\delta. \tag{70}$$

We see that the horizon radius $r_h$ is more than the hedgehog radius $\delta$:

$$r_h > \delta,$$

and our concept that "a black-hole contains the hedgehog" is justified.

## 6. Lattice-Like Structure of the False Vacuum and Non-Commutativity

Now we see, that at the Planck scale the false vacuum of the Universe is described by a non-differentiable space-time: by a foam of black-holes, having lattice-like structure, in which sites are black-holes with "hedgehog" monopoles inside them. This manifold is described by a non-commutative geometry [3,4,41,63–71].

In Refs. [3,4] B.G. Sidharth predicted:

(1)　That a cosmological constant is given by a tiny value:

$$\Lambda \sim H_0^2, \tag{71}$$

where $H_0$ is the Hubble rate in the early Universe:

$$H_0 \simeq 1.5 \times 10^{-42} \text{ GeV}. \tag{72}$$

(2)　That a Dark Energy density is very small:

$$\rho_{DE} \simeq 10^{-12} \text{ eV}^4 = 10^{-48} \text{ GeV}^4; \tag{73}$$

(3)　That a very small DE-density provides an accelerating expansion of our Universe after the Big Bang.

Sidharth proceeded from the following points of view [65]: Modern Quantum Gravity [72] (Loop Quantum Gravity, etc.,) deal with a non-differentiable space-time manifold. In such an approach, there exists a minimal space-time cut off $\lambda_{min}$, which leads to the non-commutative geometry, a feature shared by the Fuzzy Space-Time also.

If the space-time is fuzzy, non-differentiable, then it has to be described by a non-commutative geometry with the coordinates obeying the following commutation relations:

$$[dx^{\mu}, dx^{\nu}] \approx \beta^{\mu\nu} l^2 \neq 0. \tag{74}$$

Equation (74) is true for any minimal cut off $l$.

Previously the following commutation relation was considered by H.S. Snyder [73]:

$$[x, p] = \hbar \left( 1 + \left( \frac{l}{\hbar} \right)^2 p^2 \right), \ etc., \tag{75}$$

which shows that effectively 4-momentum $p$ is replaced by

$$p \rightarrow p \left( 1 + \left( \frac{l}{\hbar} \right)^2 p^2 \right)^{-1}. \tag{76}$$

Then the energy-momentum formula becomes as:

$$E^2 = m^2 + p^2 \left( 1 + \left( \frac{l}{\hbar} \right)^2 p^2 \right)^{-2}, \tag{77}$$

or

$$E^2 \approx m^2 + p^2 - 2 \left( \frac{l}{\hbar} \right)^2 p^4. \tag{78}$$

In such a theory the usual energy momentum dispersion relations are modified [66–70]. In the above equations, $l$ stands for a minimal (fundamental) length, which could be the Planck length $\lambda_{Pl}$, or for more generally—Compton wavelength $\lambda_c$.

Writing Equation (78) as

$$E = E' + E'', \tag{79}$$

where $E'$ is the usual (old) expression for energy, and $E''$ is the new additional term in modification. $E''$ can be easily verified as $E'' = -m_b c^2$—for boson fields, and $E'' = +m_f c^2$—for fermion fields with masses $m_b, m_f$, respectively. These formulas help to identify the DE density, what was first realized by B.G. Sidharth in Ref. [4].

DE density is a density of the quantum vacuum energy of the Universe. Quantum vacuum, described by Zero Point Fields (ZPF) contributions, is the lowest state of any Quantum Field Theory (QFT), and due to the Heisenberg's principle has an infinite value, which is renormalizable.

As it was pointed out in Refs. [63,74], the quantum vacuum of the Universe can be a source of the cosmic repulsion. However, a difficulty in this approach has been that the value of the cosmological constant turns out to be huge [74], far beyond the value which is observed by astrophysical measurements. This phenomenon has been called "the cosmological constant problem" [75].

A global monopole is a heavy object formed as a result of the gauge-symmetry breaking during the phase transition of the isoscalar triplet $\Phi^a$ system. The black-holes-hedgehogs are similar to elementary particles because a major part of their energy is concentrated in a small region near the monopole core. Assuming that the Planck scale false vacuum is described by a non- differentiable space-time having lattice-like structure, where sites of the lattice are black-holes with "hedgehog" monopoles inside them, we describe this manifold by a non-commutative geometry with a minimal length $l = \lambda_{Pl}$. Using the

non-commutative theory of the discrete space-time, B.G. Sidharth predicted in Refs. [4,63] a tiny value of the cosmological constant: $\Lambda \simeq 10^{-84}$ GeV$^2$ as a result of the compensation of ZPF contributions by non-commutative contributions of the hedgehog lattice.

## 7. The Phase Transition from the "False Vacuum" to the "True Vacuum"

In the Guendelman-Rabinowitz theory [58] of the universal vacua, the authors investigated the evolution of the two phases:

(1)    one being the "false vacuum" (Planck scale vacuum), and
(2)    the other—the "true vacuum" (EW-scale vacuum).

By cosmological theory, the Universe exists in the Planck scale phase for extremely short time. By this reason, the Planck scale phase was called "the false vacuum". The presence of hedgehogs as defects is responsible for the destabilization of the false vacuum. The decay of the false vacuum is accompanied by the decay of the black-holes-hedgehogs. These configurations are unstable, and at some finite cosmic temperature which is called the critical temperature $T_c$, a system exhibits a spontaneous symmetry breaking, and we observe a phase transition from the bubble with the false vacuum to the bubble with the true vacuum. After the phase transition, the Universe begins its evolution toward the low energy Electroweak (EW) phase. Here the Universe underwent the inflation, which led to the phase having the VEV $v_1 \approx 246$ GeV. This is a "true" vacuum, in which we live.

Guendelman and Rabinowitz [58] also allowed a possibility to consider an arbitrary domain wall between these two phases. During the inflation, domain wall annihilates, producing gravitational waves and a lot of the SM particles, having masses.

The Electroweak spontaneous breakdown of symmetry $SU(2)_L \times U(1)_Y \to U(1)_{el.mag}$ leads to the creation of the topological defects in the EW vacuum. They are the Abrikosov-Nielsen-Olesen closed magnetic vortices ("ANO strings") of the Abelian Higgs model [76,77], and Sidharth's Compton phase objects [78–80]. Then the electroweak vacuum again presents the non-differentiable manifold, and we have to consider the non-commutative geometry.

Kirzhnits [81] and Linde [82] were first who considered the analogy between the Higgs mechanism and superconductivity, and argued that the SM ($SU(2)$-doublet) Higgs field condensate $v_1 = \langle H \rangle \approx$ 246 GeV disappears at high temperatures, leading to the symmetry restoration. As a result, at high temperatures $T > T_c$ all fermions and bosons are massless. These conclusions were confirmed, and the critical temperature was estimated (see review by A. Linde [83]).

At the early stage, the Universe was very hot, but then it began to cool down. Black-holes-monopoles (as bubbles of the vapour in the boiling water) began to disappear. The temperature dependent part of the energy density died away. In that case, only the vacuum energy will survive. Since this is a constant, the Universe expands exponentially, and an exponentially expanding Universe leads to the inflation (see review [84]). While the Universe was expanding exponentially, so it was cooling exponentially. This scenario was called supercooling in the false vacuum. When the temperature reached the critical value $T_c$, the Higgs mechanism of the SM created a new condensate $\phi_{min1}$, and the vacuum became similar to a superconductor, in which the topological defects are magnetic vortices. The energy of black-holes is released as particles, which were created during the radiation era of the Universe, and all these particles (quarks, leptons, vector bosons) acquired their masses $m_i$ through the Yukawa coupling mechanism $Y_f \bar{\psi}_f \psi_f \phi$. Therefore, they acquired the Compton wavelength, $\lambda_i = \hbar / m_i c$. Then according to the Sidharth's theory of the cosmological constant, in the EW-vacuum we again have lattice-like structures formed by bosons and fermions, and the lattice parameters "$l_i$" are equal to the Compton wavelengths: $l_i = \lambda_i = \hbar / m_i c$.

As it was shown in Ref. [41], the Planck scale vacuum energy density (with the VEV $v_2$) is equal to:

$$\rho_{vac}(\text{at Planck scale}) = \rho_{ZPF}(\text{at Planck scale}) - \rho_{black\ holes}^{(NC)} \approx 0, \tag{80}$$

and the EW-vacuum gives:

$$\rho_{vac}(\text{at EW scale}) =$$
$$\rho_{ZPF}(\text{at EW scale}) - \rho_{vortex\ contr.}^{(NC)} - \rho_{boson\ fields}^{(NC)} + \rho_{fermion\ fields}^{(NC)} \approx 0. \tag{81}$$

In the above equations "NC" means the "non-commutativity" and "ZPF" means "zero point fields".

Assuming by example that hedgehogs form a hypercubic lattice with lattice parameter $l = \lambda_{Pl}$, we have the negative energy density of such a lattice equal to:

$$\rho_{lat} \simeq -M_{BH}M_{Pl}^3. \tag{82}$$

If this energy density of the hedgehogs lattice compensates the Einstein's vacuum energy (44), we have the following equation:

$$\frac{\lambda}{4}v^4 \approx |M_{BH}|M_{Pl}^3, \tag{83}$$

Using the estimation (37), we obtain:

$$\frac{3}{2}M_{Pl}^4 \approx |M_{BH}|M_{Pl}^3, \tag{84}$$

or

$$|M_{BH}| = \frac{3}{2}M_{Pl} \approx 3.65 \times 10^{18} \text{ GeV}. \tag{85}$$

Therefore hedgehogs have a huge mass of order of the Planck mass. Equation (65) predicts a radius $\delta$ of the hedgehog's core:

$$\delta \approx \frac{|M_{BH}|}{8\pi v^2} \approx \left(\frac{128\pi}{3}M_{Pl}\right)^{-1} \sim 10^{-21} \text{ GeV}^{-1}. \tag{86}$$

### 7.1. Stability of the EW Vacuum

Here we emphasize that due to the energy conservation law, the vacuum density before the phase transition (for $T > T_c$) is equal to the vacuum density after the phase transition (for $T < T_c$), therefore we have:

$$\rho_{vac}(\text{at Planck scale}) = \rho_{vac}(\text{at EW scale}). \tag{87}$$

The analogous link between the Planck scale phase and EW phase was considered in the paper [78]. It was shown that the vacuum energy density (DE) is described by the different contributions to the Planck and EW scale phases. This difference is a result of the phase transition. However, the vacuum energy densities (DE) of both vacua are equal, and we have a link between gravitation and electromagnetism via the Dark Energy. According to the last equation (87), we see that if $\rho_{vac}$ (at the Planck scale) is almost zero, then $\rho_{vac}$ (at EW scale) also is almost zero, and we have a triumph of the Multiple Point Principle: we have two degenerate vacua with almost zero vacuum energy density. Almost zero cosmological constants are equal:

$$\Lambda_1 = \Lambda_2 \approx 0,$$

where $\Lambda_i$ is a cosmological constant for $i$-vacuum with VEV $v_i$ (here $i = 1, 2$).

Now we see that we have obtained a very important result: our vacuum, in which we live, is stable. The Planck scale vacuum cannot be negative: $V_{eff}(min_1) = V_{eff}(min_2)$ exactly.

## 8. Hedgehogs in the Wilson Loops and the Phase Transition in the $SU(2)$ Yang-Mills Theory

The authors of Ref. [13] investigated the gauge-invariant hedgehog-like structures in the Wilson loops of the $SU(2)$ Yang-Mills theory. In this model the triplet Higgs field $\hat{\Phi} = \frac{1}{2}\Phi^a\sigma^a$ vanishes at the centre of the monopole $x = x_0$: $\Phi(x_0) = 0$, and has a generic hedgehog structure in the spatial vicinity of this monopole.

In the Yang-Mills theory, a hedgehog structure can be entirely defined in terms of Wilson-loop variables [14]. In general, we consider an untraced Wilson loop, beginning and ending at the point $x_0$ on the closed loop $C$:

$$W_C(x_0) = P\exp ig \oint_C dx_\mu \hat{A}_\mu. \tag{88}$$

To improve the analogy with the triplet Higgs field $\hat{\Phi}$, we subtract the singlet part from $W_C(x_0)$:

$$\hat{\Gamma}_C(x_0) = W_C(x_0) - 1 \cdot \frac{1}{2}TrW_C(x_0). \tag{89}$$

This is a traceless adjoint operator similar to the field $\hat{\Phi}$. Associating the triplet part $\hat{\Gamma}_C(x_0)$ of Wilson loop $W_C(x_0)$ with the triplet Higgs field $\hat{\Phi}$, we notice the following property: As the Higgs field vanishes in all points $x$, belonging to the monopole trajectory, similarly $\Gamma_C$ vanishes on the hedgehog loop $C$:

$$W_C \in Z_2 \Leftrightarrow \Gamma_C = 0.$$

In conventional superconductivity [76], Abrikosov vortices are singularities in the superconducting condensate (i.e., in the Cooper-pair field). Abrikosov vortices are "two-dimensional hedgehogs" (see Ref. [58]). In the core of the Abrikosov's vortices, the superconductivity is broken, and the normal state is restored. As temperature increases, the condensate weakens, and nucleation of the vortices due to thermal fluctuations strengthens. Thus, the higher the temperature is, the density of the (thermal) vortices should be larger. It can be expected in the YM theory that the density of hedgehog loops is also sensitive to the phase transition.

The order parameter of the phase transition is the vacuum expectation value (trace) of the Polyakov line:

$$\hat{L}(x) = P\exp ig \int_0^{1/T} dx_4 A_4(\vec{x}, x_4). \tag{90}$$

Here $T$ is a temperature and VEV is $L = \frac{1}{2}Tr\hat{L}$. Functional $\hat{L}(x)$, called the thermal Wilson line, is a basic variable in an effective theory, which describes the properties of the finite-temperature phase transition of the system. In the confinement phase, the expectation value of the Polyakov line is zero: $\langle L \rangle = e^{-TF_q} = 0$, indicating that the free energy of a single quark becomes infinite when $F_q \to \infty$. In the deconfinement phase, the Polyakov line has a non-zero expectation value: $\langle L \rangle \neq 0$, and the quarks are no longer confined. Considering lattice model of the $SU(2)$ Yang-Mills theory, Belavin, Chernodub and Kozlov showed numerically that the density of hedgehogs structures in the thermal Wilson-Polyakov lines is very sensitive to the finite-temperature phase transition. The hedgehog line density behaves like an order parameter: the density is almost independent of the temperature in the confinement phase and changes substantially as the system enters the deconfinement phase. These authors obtained a very important result: $\beta_{crit} \approx 2.5$, which shows that the critical temperature $T_c$, corresponding to the hedgehogs' confinement, is smaller than the Planck scale value.

Indeed,

$$\beta = 1/g^2 = 1/(4\pi\alpha) = \frac{1}{T\lambda_{Pl}}. \tag{91}$$

Then the critical temperature is:

$$T_c = \frac{M_{Pl}}{\beta_{crit}} \approx 0.4M_{Pl} \approx 10^{18} \text{ GeV}. \tag{92}$$

## 9. Threshold Energy of the $SU(2)$-Triplet Higgs Bosons

Equation (91) also gives the critical value of the couplingconstant $g^2_{crit}$ of the $SU(2)$ Yang-Mills theory:

$$g^2_{crit} \approx 0.4, \tag{93}$$

or

$$\alpha^{-1}_{crit} \approx 4\pi \times 2.5 \approx 31.4. \tag{94}$$

The renormalization group equation (RGE) for $\alpha^{-1}(\mu)$ (see for example [85] and references there) is given by the following expression at the one-loop level:

$$\alpha^{-1}(\mu) = \alpha(M_t)^{-1} + bt, \tag{95}$$

where $t = \ln(\mu/M_t)$, and $M_t \simeq 173.34$ GeV is the top quark mass.

Usually RGE is a function of $x$: $x = \log_{10} \mu$. Then

$$t = \ln\left(\frac{10^x}{M_t}\right) = x \ln 10 - \ln M_t \approx 2.30x - 5.16. \tag{96}$$

For $SU(2)$-gauge theory $b \approx 19/12\pi$ and $\alpha^{-1}_2(M_t) \approx 29.4 \pm 0.02$, and we obtain the following RGE equation [85]:

$$\alpha^{-1}_2(x) \approx 29.4 + 0.504(2.30x - 5.16). \tag{97}$$

Then we can calculate $x_{crit}$ using the following result:

$$\alpha^{-1}_{crit} \approx 31.4 = 29.4 + 1.16x_{crit} - 2.60, \tag{98}$$

which gives:

$$x_{crit} \sim 4,$$

or

$$\mu_{crit} \sim 10^4 \text{ GeV}.$$

This result means that the hedgehog's confinement happens at energy of 10 TeV, which is a threshold energy of the production of a pair of the $SU(2)$-triplet Higgs bosons $\Phi^a$:

$$E_{threshold} \sim 10^4 \text{ GeV} = 10 \text{ TeV}. \tag{99}$$

At this energy we can expect to see at LHC the production of the triplet Higgs particles with mass $\sim$5 TeV. If we assume that in the region $E > E_{threshold}$ the effective Higgs potential has an interaction between the triplet field $\Phi^a$ and Higgs doublet $H^\alpha$ (here $a = 1, 2, 3$ and $\alpha = 1, 2$), then we have such an effective Higgs potential with two Higgs fields: $SU(2)$-triplet $\Phi^a$ and $SU(2)$-doublet $H$:

$$\begin{aligned} V_{eff} &= \lambda_{h,eff}(h)\left(|\Phi|^2 - v^2_2\right)^2 + \lambda_{H,eff}(H)\left(|H|^2 - v^2_1\right)^2 \\ &+ \lambda_{hH,eff}(h, H)\left(|\Phi|^2 - v^2_2\right)\left(|H|^2 - v^2_1\right) + \Lambda. \end{aligned} \tag{100}$$

At $T = T_c$, we have the phase transition in the Universe when the electroweak spontaneous breakdown of symmetry $SU(2)_L \times U(1)_Y \rightarrow U(1)_{el.mag}$ creates new topological defects of the EW vacuum: the Abrikosov-Nielsen-Olesen closed magnetic vortices ("ANO strings") of an Abelian Higgs model [76,77] and point-like Compton phase objects [78–80]. Therefore below energy $E = E_{threshold}$ we have the following effective Higgs potential:

$$V^{(1)}_{eff} = \lambda_{H,eff}(H)\left(|H|^2 - v^2_1\right)^2 + \Lambda, \tag{101}$$

which has the low-energy first vacuum with the VEV $v_1$.

Here it is necessary to comment that our Gravi-Weak unification described in Section 4 is not valid exactly due to the presence of a mixing term in the effective Higgs potential $V_{eff}$. This unification is not correct if the mixing coupling constant $\lambda_{hH,eff}$ is not very small and negligible. The hedgehog's parameters obtained in Sections 4 and 5 are approximately valid if $\lambda_{hH,eff} \ll 1$. In this paper, we assume that this coupling $\lambda_{hH,eff}$ is negligibly small.

A cosmological constant $\Lambda$ in Equations (100) and (101) is given by the tiny value of DE (see Equation (3)).

## 10. The Higgs Mass and Vacuum Stability/Metastability in the Standard Model

As it was mentioned in Section 2, assuming the existence of two degenerate vacua in the SM (the first Electroweak vacuum and the second Planck scale one), Froggatt and Nielsen predicted the top-quark and Higgs boson masses: $M_t = 173 \pm 5$ GeV and $M_H = 135 \pm 10$ GeV [15]. Their prediction for the mass of the SM $SU(2)$-doublet Higgs boson was improved in Ref. [86] by calculations of the two-loop radiative corrections to the effective Higgs potential $V_{eff}(H)$ (here $H^2 \equiv \phi^\dagger \phi$). The prediction of Ref. [86]: $M_H = 129 \pm 2$ GeV provided the possibility of the theoretical explanation of the value $M_H \simeq 125.7$ GeV observed at LHC.

The authors of reference [87] extrapolated the SM parameters up to the high (Planck) energies with full 3-loop NNLO RGE precision. From Degrassi et al. calculation [86], the effective Higgs field potential $V_{eff}(H)$ has a minimum, which slightly goes under zero, so that the present EW-vacuum is unstable for the experimental Higgs mass $M_H \simeq 125.09 \pm 0.24$ GeV, while the value that would have made the second minimum $v_2$ just degenerate with the present vacuum $v_1$ would be rather $m_H \simeq 129.4$ GeV.

A theory of a single scalar field is given by the effective potential $V_{eff}(\phi_c)$ which is a function of the classical field $\phi_c$. In the loop expansion $V_{eff}$ is given by a series:

$$V_{eff} = V(0) + \Sigma_{n=1} V^{(n)}, \tag{102}$$

where $V(0)$ is the tree-level potential of the SM:

$$V(0) = -\frac{1}{2} m_H^2 \phi^2 + \frac{1}{4} \lambda_H \phi^4. \tag{103}$$

The vast majority of the available experimental data is consistent with the SM predictions. No sign of new physics has been detected. Until now there is no evidence for the existence of any particles other than those of the SM, or bound states composed of other particles. All accelerator physics seems to fit well with the SM, except for neutrino oscillations. These results caused a keen interest in the possibility of the emergence of new physics only at very high (Planck scale) energies and generated a great attention to the problem of the vacuum stability: whether the EW-vacuum is stable, unstable, or metastable. A largely explored scenarios assume that new physics comes only at the Planck scale $M_{Pl} = 1.22 \times 10^{19}$ GeV. According to these scenarios, we need the knowledge of the Higgs effective potential $V_{eff}(\phi)$ at very high values of $\phi$.

The loop corrections give the $V_{eff}$ with values of $\phi$, which are much larger than $v_1 \approx 246$ GeV. The effective Higgs potential develops a new minimum at $v_2 \gg v_1$. The position of the second minimum depends on the SM parameters, especially on the top and Higgs masses, $M_t$ and $M_H$. This $V_{eff}(min2)$ can be higher or lower than the $V_{eff}(min1)$ showing a stable EW vacuum (in the first case), or metastable one (in the second case). The red solid line of Figure 2 by Degrassi et al. shows the running of the $\lambda_{H,eff}(\phi)$ for $M_H \simeq 125.7$ GeV and $M_t \simeq 171.43$ GeV, which just corresponds to the

stability line, that is, to the stable EW-vacuum. In this case the minimum of the $V_{eff}(H)$ exists at the $\phi = \phi_0 \sim 10^{18}$ GeV, where according to MPP:

$$\lambda_{H,eff}(\phi_0) = \beta(\lambda_{H,eff}(\phi_0)) = 0.$$

Unfortunately, according to Refs. [86,87], this case does not correspond to the current experimental values.

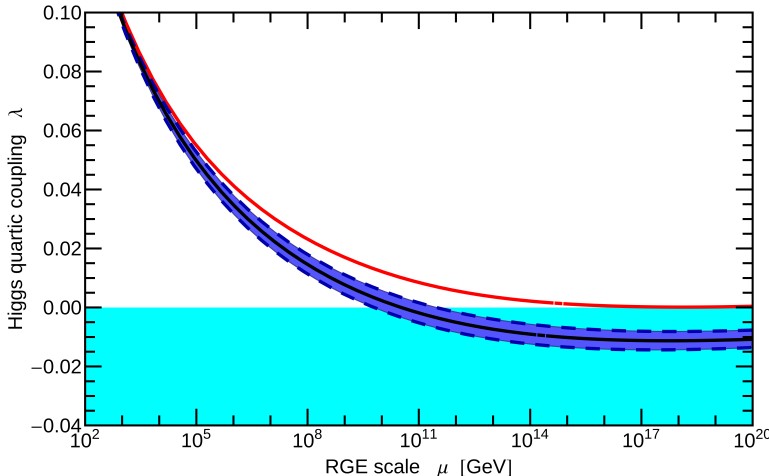

**Figure 2.** The renormalization group (RG) evolution of the Higgs selfcoupling $\lambda$ for $M_t \simeq 173.34$ GeV and $\alpha_s = 0.1184$ given by $\pm 3\sigma$. Blue lines present metastability for current experimental data, red (thick) line corresponds to the stability of the EW vacuum.

In Figure 2 blue lines (thick and dashed) present the RG evolution of $\lambda_H(\mu)$ for current experimental values $M_H \simeq 125.7$ GeV and $M_t \simeq 173.34$ GeV. The thick blue line corresponds to the central value of $\alpha_s = 0.1184$ and dashed blue lines correspond to its errors equal to $\pm 0.0007$. We see that absolute stability of the Higgs potential is excluded by at 98% C.L. for $M_H < 126$ GeV. Figure 2 shows that asymptotically $\lambda_H(\mu)$ does not reach zero but approaches to the negative value:

$$\lambda_H \to -0.01 \pm 0.002, \tag{104}$$

indicating the metastability of the EW vacuum. According to the paper [86], the stability line is given in Figure 2 by the red thick line and corresponds to $M_H = 129.4 \pm 1.8$ GeV. We see that the current experimental values of $M_H$ and $M_t$ show the metastability of the present EW-vacuum of the Universe, and this result means that the MPP law is not exact.

## 11. A New Physics in the SM

Can the MPP be exact due to the corrections from hedgehogs' contributions? We think that it is possible.

If we assume that in the region $E > E_{threshold}$ the effective Higgs potential contains not only the $SU(2)$-triplet field $\Phi^a$, but also the $SU(2)$-doublet Higgs field $H^\alpha$ (where $a = 1, 2, 3$ and $\alpha = 1, 2$), then there exists an interaction (mixing term) between these two Higgs fields as it was shown in Equation (100). Of course, the effective Higgs self-interaction coupling constant $\lambda_{H,eff}(\mu)$ is a running function presenting loop corrections to the Higgs mass $M_H$, which arise from the Higgs bosons $H$ ($\Delta\lambda_H(\mu)$) and from hedgehogs $h$ ($\delta\lambda_H(\mu)$):

$$\lambda_{H,eff}(\mu) = \frac{G_F}{\sqrt{2}} M_H^2 + \Delta\lambda_H(\mu) + \delta\lambda_H(\mu), \tag{105}$$

where $G_F$ is the Fermi constant. The main contribution to the correction $\delta\lambda_H(\mu)$, described by a series in the mixing coupling constant $\lambda_{hH}$, is a term $\lambda_S$ given by the Feynman diagram of Figure 3 containing the hedgehog $h$ in the loop:

$$\delta\lambda_H(\mu) = \Sigma_n c_n(\mu)\lambda_{hH}^{2n} = \lambda_S(\mu) + .... \tag{106}$$

Here the effective Higgs self-interaction coupling constant $\lambda_{H,eff}(\mu)$ is equal to $\lambda_{eff}(\mu)$ considered in Refs. [86,87].

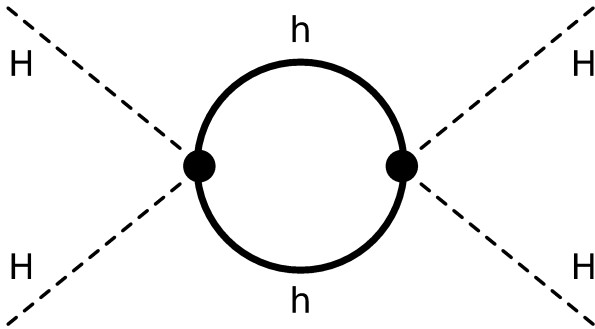

**Figure 3.** The main Feynman diagram containing hedgehogs in the loop, which corrects the effective Higgs mass.

Our hedgehog is an extended object with a mass $M_h$ and radius $R_h$, therefore it is easy to estimate $\lambda_S$ at high energies $\mu > E_{threshold}$ by methods of Ref. [33]:

$$\lambda_S(\mu) \approx \frac{1}{16\pi^2}\frac{\lambda_{hH}^2(\mu)}{(R_h M_h)^4}, \tag{107}$$

where $\lambda_{hH}(\mu)$ is a running coupling constant of the interaction of hedgehogs $h$ with the Higgs fields $H$ (see Equation (100)). In Equation (32) parameters $M_h = |M_{BH}|$ and $R_h$ are the running mass and radius of the hedgehog, respectively. According to Equations (65), (85) and (86), we have:

$$M_h(\mu) = 8\pi v^2 \delta(\mu) \quad \text{and} \quad R_h(\mu) = \delta(\mu). \tag{108}$$

At high Planck scale energies, they are:

$$M_h \sim 10^{18} \text{ GeV}, \quad R_h \sim 10^{-21} \text{ GeV}^{-1}, \tag{109}$$

and

$$R_h M_h \sim 10^{-3}. \tag{110}$$

As a result, asymptotically we have:

$$\lambda_S \sim \frac{\lambda_{hH}^2}{16\pi^2}10^{12}. \tag{111}$$

If hedgehog parameter $\lambda_{hH}$ is:

$$\lambda_{hH} \sim 10^{-6}, \tag{112}$$

then

$$\lambda_S \sim 0.01, \tag{113}$$

and the hedgehogs' contribution transforms the metastable (blue) curve of Figure 2 into the stable (red) curve, and we have an exact stability of the EW-vacuum and the accuracy of the MPP with two degenerate vacua in the Universe.

A tiny value of the mixing coupling $\lambda_{hH}$, given by Equation (112), confirms a good accuracy of our calculations in the framework of the GWU model. Of course, the results obtained in our investigation

depend on details of the $f(R)$ gravity and Gravi-Weak unification model. Nevertheless, we predict a production of triplet Higgs bosons at LHC at energy scale ~10 TeV and the existence of two degenerate, or almost degenerate vacua of our Universe provided by the existence of black-holes-hedgehogs in the false Planck scale vacuum.

## 12. Conclusions

(1)   In this investigation, we have based on the discovery that a cosmological constant of our Universe is extremely small, almost zero, and assumed a new law of Nature which was named as a Multiple Point Principle (MPP). The MPP postulates: There are two vacua in the SM with the same energy density, or cosmological constant, and both cosmological constants are zero, or approximately zero. We considered the existence of the following two degenerate vacua in the SM: (a) the first Electroweak vacuum at $v_1 = 246$ GeV, which is a "true" vacuum, and (b) the second "false" vacuum at the Planck scale with VEV $v_2 \sim 10^{18}$ GeV.

(2)   The bubble, which we refer to as "the false vacuum", is a de-Sitter space with its constant expansion rate $H_F$. The initial radius of this bubble is close to the de-Sitter horizon, which corresponds to the Universe radius. The space-time inside the bubble, which we refer to as "the true vacuum", has the geometry of an open FLRW Universe.

(3)   We investigated the topological structure of the universal vacua. Different phase transitions, which were resulted during the expansion of the early Universe after the Planck era, produced the formation of the various kind of topological defects. The aim of this investigation is the consideration of the hedgehog configurations as defects in the false vacuum. We have obtained a solution for a black-hole in the region which contains a global monopole in the framework of the $f(R)$ gravity, where $f(R)$ is a function of the Ricci scalar $R$. Here we have used the results of the Gravi-Weak unification (GWU) model. The gravitational field, isovector scalar $\Phi^a$ with $a = 1, 2, 3$, produced by a spherically symmetric configuration in the scalar field theory, is pointing radially: $\Phi^a$ is parallel to $\hat{r}$—the unit vector in the radial direction. In this GWU approach, we obtained a "hedgehog" solution (in Alexander Polyakov's terminology). We also showed that this is a black-hole solution, corresponding to a global monopole that has been "swallowed" by a black-hole.

(4)   We estimated all parameters of the Gravi-Weak unification model, which gave the prediction of the Planck scale false vacuum VEV equal to $v = 2\sqrt{2} M_{Pl}^{red} \approx 6.28 \times 10^{18}$ GeV.

(5)   We have shown, that the Planck scale Universe vacuum is described by a non-differentiable space-time: by a foam of black-holes, or by lattice-like structure, where sites are black-holes with the "hedgehog" monopoles inside them. This manifold is described by a non-commutative geometry, leading to a tiny value of cosmological constant $\Lambda \approx 0$.

(6)   Taking into account that the phase transition from the "false vacuum" to the "true vacuum" is a consequence of the electroweak spontaneous breakdown of symmetry $SU(2)_L \times U(1)_Y \rightarrow U(1)_{el.mag}$, we considered topological defects of EW-vacuum: the Abrikosov-Nielsen-Olesen closed magnetic vortices ("ANO strings") of the Abelian Higgs model and Sidharth's Compton phase objects. We showed that the "true vacuum" (EW-vacuum) again is presented by the non-differentiable manifold with non-commutative geometry leading to an almost zero cosmological constant.

(7)   By solving the gravitational field equations we estimated the black hole-hedgehog's mass, radius and horizon radius are $M_h \approx 3.65 \times 10^{18}$ GeV, $R_h \sim 10^{-21}$ GeV$^{-1}$ and $r_h \approx 2.29 R_h$ respectively.

(8)   We considered that due to the energy conservation law, the vacuum energy density before the phase transition is equal to the vacuum energy density after the phase transition: $\rho_{vac}$(at Planck scale) $= \rho_{vac}$(at EW scale). This result confirms the Multiple Point Principle: we have two degenerate vacua $v_1$ and $v_2$ with an almost zero vacuum energy density (cosmological constants). By these considerations, we confirmed the vacuum stability of the

EW-vacuum, in which we live. The Planck scale vacuum cannot be negative because of the exact equality $V_{eff}(min_1) = V_{eff}(min_2)$.

(9) Hedgehogs in the Wilson loops of the $SU(2)$ Yang-Mills theory, and phase transitions in this theory were investigated revising the results of Refs. [13,14]. Using their lattice result for the critical value of the temperature of hedgehog's confinement phase: $\beta_{crit} \approx 2.5$, we predicted the production of the $SU(2)$-triplet Higgs bosons at LHC at energy scale $\mu \sim 10$ TeV, providing a new physics in the SM.

(10) We considered an additional confirmation of the vacuum stability and accuracy of the MPP taking into account that hedgehog fields $\Phi^a$ produce a new physics at the scale $\sim 10$ TeV, and calculating at high energies the contribution of the black-hole-hedgehog corrections to the effective Higgs potential. This result essentially depends on the hedgehog field parameters: mass, radius and mixing coupling constant $\lambda_{hH}$ of the interaction of hedgehogs with the SM doublet Higgs fields $H$.

**Funding:** This research received no external funding.

**Acknowledgments:** L.V.L. greatly thanks to the B.M. Birla Science Centre (Hyderabad, India) and personally B.G. Sidharth, for hospitality, collaboration and financial support. H.B.N. wishes to thank the Niels Bohr Institute for the status of professor emeritus and corresponding support. C.R.D. is thankful to BLTP Director D.I. Kazakov for support.

**Conflicts of Interest:** The authors declare no conflict of interest.

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
