# Peer review of "Explaining Defects of the Universal Vacua with Black Holes-Hedgehogs and Strings"

_universe, doi:10.3390/universe5030078_

Round 1
Reviewer 1 Report
I have no criticism regarding the manuscript
Author Response
We are thankful to Reviewer 1 because she/he has no criticism regarding the manuscript. We will do a minor spell check before final submission.
Reviewer 2 Report
This paper explore the consequences of the multiple point princeple, it has very interesting consequences on the stability of our vacuum and seems consisten with observations, paper should be accepted, the author may consider citing
Is Our Vacuum Stable? |
where a related discussion is made
Author Response
We are very much thankful to Reviewer 2 for her/his positive report regarding our manuscript. We will cite the article suggested by her/him in the final version.
Reviewer 3 Report
In this manuscript, the authors review different aspects associated with topological configurations within the so-called the Multiple Point Principle (MPP). I have read carefully the work and I have several questions before accepting its publication:1.- Before Eq. (11), it is claimed that the initial radii of the walls and strings are close to the de-Sitter horizon. It is not clear to me at what time does it happen.
2.- Eq. (15) have curvature square terms. I suppose they are neglected in the entire analysis, but I have not found any comment about it.It needs to be clarified.
3.- If we neglect the curvature square term, the theory is not a f(R)model. Lagrangian given by Eq. (18) does not define a f(R) model. It should be rewritten.
4.- The explicit expressions given for the Renormalization Group Equations (RGEs), I suppose are at 1-loop, as for example, Eq. (96). It should be clarified at what loop order they are working.
I would need to know these details before considering the work for publication.
Author Response
We are very much thankful to Reviewer 3 for her/his report regarding our manuscript. We will change the final version as per her/his report. We will try to do moderate English changes.
Answers to all 4 comments:
1.- The De Sitter radius is given of 10^{28} cm is like the radius of the visible part of the Universe today. The topological structure has the size of the De Sitter horizon at the time when such a topological structure may give rise toa new Universe. Then you might suspect that both the size of the structure creating so to speak the new Universe and the horizon at that starting time are of the same order. This, of course, is only a true guess if we indeed have such a proliferating Universe model in our mind. We liked such Universe proliferating model, and it is correct by introducing the words ``The initial radii ...'' two lines above Eq. (11). We have changed a little bit the sentence before Eq. (11):
"At the moment of creation of the new Universe by the new vacuum or topological structure giving rise to the initial radii of walls and strings are close to the de-Sitter horizon. This horizon corresponds to a radius today of the order:"
2.- Eq. (15) is a special case of the f(R) gravity, in Ref. [50].
3.- So, the Lagrangian given by Eq. (18) does define a f(R) model.
4.- Yes Eq. (96) is at the one-loop level. We have introduced four words "at the one-loop level:" just before Eq. (96).
Round 2
Reviewer 3 Report
The authors have not modified enough the manuscript in order to address my criticisms. I have read carefully their short answerand I do not agree with the reply of points 2 and 3 of my first report:
2.- Eq. (15) is not a f(R) model and it is not special case of the models studied in Ref. [50]. It is evident.
3.- Eq. (18) does not define a f(R) model. A f(R) model means that f(R) only depends on the scalar curvature, not on an additional scalar field.
These statements do not invalidate the analysis, but they should be corrected if the manuscript is finally published.
Author Response
We are very much thankful to reviewer 3, for her/his constructive and positive comments for the second time.
2.- She/He is right, Eq. (15) is not a f(R) model and it is not a special case of the models studied in Ref. [50-52]. We have checked it carefully now.
3.- We removed the wrong equation "Eq. (18)" which does not define a f(R), model. Also, we modified the next line with "Some special case of the f(R) gravity studied in [50–52], but in a general case ...". So that [50–52] will not be removed from the reference list.